# Ultrasensitive reversible chromophore reaction of BODIPY functions as high ratio double turn on probe

Dehui Hu[1], Tao Zhang[1], Shayu Li[1], Tianjun Yu[2], Xiaohui Zhang[2], Rui Hu[1], Jiao Feng[1], Shuangqing Wang[1], Tongling Liang[1], Jianming Chen[1], Lyubov N. Sobenina[3], Boris A. Trofimov[3], Yi Li[2], Jinshi Ma[1] & Guoqiang Yang[1]

Chromophore reactions with changes to conjugation degree, especially those between the conjugated and unconjugated state, will bring a large spectral variation. To realize such a process, a meso-naked BODIPY (**MNBOD**) with two electron-withdrawing groups around the core is designed and synthesized. The resulting system is extremely sensitive to bases. The red, highly fluorescent solution readily becomes colorless and non-fluorescent after base addition; however, the color and fluorescence can be totally and instantly restored by addition of acid or formaldehyde. Analyses show that two identical **MNBOD**s are connected by a C–C single bond ($sp^3$) at the meso-position through a radical reaction that results in an unconjugated, colorless dimer complexed with bases. When the bases are consumed, the dimer immediately dissociates into the red, highly fluorescent, conjugated **MNBOD** monomer. With 260 nm spectral change and over 120,000 turn-on ratio, this chromophore-reaction can be utilized as a sensitive reaction-based dual-signal probe.

---

[1] Beijing National Laboratory for Molecular Sciences, Key Laboratory of Photochemistry, Institute of Chemistry, University of Chinese Academy of Sciences, Chinese Academy of Sciences, Beijing 100190, China. [2] Technical Institute of Physics & Chemistry, Key Laboratory of Photochemical Conversion & Optoelectronic Materials, University of Chinese Academy of Sciences, Chinese Academy of Sciences, Beijing 100190, China. [3] A.E. Favorsky Irkutsk Institute of Chemistry, Siberian Branch of the Russian Academy of Sciences, Irkutsk 664033, Russia. Correspondence and requests for materials should be addressed to D.H. (email: jobs@hudehui.com) or to B.A.T. (email: boris_trofimov@irioch.irk.ru) or to G.Y. (email: gqyang@iccas.ac.cn)

Fluorescent probes that make an analyte visible and distinguishable are of great importance for sensing and imaging. A variety of strategies[1,2] for the development of fluorescent probes have been being pursued. Most strategies are based on photophysical processes such as PET (photoinduced electron transfer), ICT (intramolecular charge transfer), FRET (fluorescence Förster resonance energy transfer), and AIE (aggregation-induced emission)[3], while few are based on chemical reactions. Reaction-based probes have attracted considerable attention in recent years due to their high selectivity and sensitivity, especially for ion recognition and bioimaging[4,5]. Chang[6,7], Sessler[8] and Peng[9] et al. have reviewed these reaction-based probes. A conclusion that can be drawn from these reviews is that most reaction-based probes utilize photophysical processes, most notably PET; for instance, a specific chemical reaction to interrupt a pre-quenching PET process and revival of the fluorescent emission[10,11]. Few probes[2,12–18] are, however, based on direct chromophore conjugation change (i.e., chromophore-reaction)[9].

The turn-on process is a universal methodology for all strategies aforementioned, for which the low background and high turn-on ratio are of critical importance in terms of probe sensitivity[5]. With the change of chromophore structure together with conjugation degree, the chromophore-reaction-based probes could have an advantage over the photophysical ones with much larger spectrum shift. The possible low background (ideally zero, non-conjugated state) and high turn-on state (conjugated state) will obviously benefit the turn-on process. Maximal output vs. minimal background will probably induce an ultrahigh turn-on ratio, which is quite essential for fluorescent sensing and imaging, e.g., trace detection and single-molecule fluorescence-based super-resolution microscopy techniques[19,20]. Moreover, the structure change of chromophore may induce a dual signal, double turn-on process with synchronous colorimetric and fluorescence change, which will benefit the visualization process and precision.

The versatile BODIPY (boron dipyrromethene) dyes with strong absorption and high fluorescent emission in the visible region[21] have been widely used as fluorescent probes for their excellent properties[3,21–25]. However, with its generally accepted chemical robustness and environmental insensitivity, there is no report on using the structure change of the BODIPY chromophore as a chemical probe.

Here we report a new type of BODIPY and also a molecular strategy that can realize the reversible transformation between BODIPY and dipyrrolmethane by chromophore-reaction. That transformation results in an alteration between nearly zero background state and a strong, dual signal state. The spectrum shift even up to 260 nm and the turn-on ratio can reach up to 120,000. To the best of our knowledge, there's no report of photophysical process has such sharp spectra change. In addition, we show the potential to achieve the detection of formaldehyde and temperature by that BODIPY chromophore-reaction process.

## Results

**Molecular design.** The BODIPY has both strong absorption and fluorescence in the visible region; on the contrary the dipyrrolmethane has neighter absorption nor fluorescence in that region[26,27]. That difference in structure and great distinction in photophysical properties inspired us to find means to realize the derivative or vice versa. As hypothesized in Fig. 1, a transformation between BODIPY and boron dipyrrolmethane could obviously minimize the background and maximize the report signal with a "dual-signal", "double turn-on" process. However, to realize such transformation, we should get rid of the doctrine that the BODIPY is chemical robustness and environmental insensitivity.

As we learned that the *meso*-position of BODIPY is more electron deficient than other sites[28], we designed and synthesized a *meso*-naked BODIPY (**MNBOD**, **1**) with electron-withdrawing groups (EWGs) connected to the BODIPY core (Fig. 2a). We expected that the EWG modification will enhance the electron deficiency of the *meso*-position, thus making it easily reduced or activated as a proton donor. This new type of BODIPY could be transformed into boron dipyrrolmethane as hypothesized in Fig. 1 and be utilized as a chromophore reaction-based "dual-signal", "high turn-on ratio" fluorescent probe.

**Phenomena and characterization.** The **MNBOD 1** solution underwent a fast reaction with such bases as $NH_2NH_2$ and $NH_2CH_2CH_2NH_2$. Instantly after addition of bases, the red color accompanying the characteristic fluorescence vanished without a trace. The colorless system could be restored totally to red after addition of organic/inorganic acid, as shown in Fig. 2b. Furthermore, the vanishing and reversion of the characteristic absorption (546 nm) and fluorescence emission (564 nm) of **1** could be cycled; one example of $NH_2NH_2$/ethanol and HCl/ethanol mediation is depicted in Fig. 2c. Another example of NaOH and HAc (acetic acid) is depicted in Supplementary Fig. 1.

The cycle number was tested by UV absorption in a 10 mm × 10 mm quartz cell with 2 mL of $10^{-4}$ M **MNBOD** ethanol solution (Fig. 2d). The 0.1 M of NaOH ethanol solution and 1 M of HCl aqueous solution were used as base and acid. The volume of base and acid added and the subsequent UV absorption at 276 nm and 536 nm for each cycle were listed in Supplementary Table 1.

From Supplementary Table 1, we can see an average absorption decay of 0.025 (i.e., $OD_{A(n+1)}/OD_{An} = 0.975$) for each cycle, which means the absorption at 536 nm will decay to a half initial value at cycle 27. The solution volume will notably increase as cycle number increasing, which decreases the concentration and subsequently decreases the maximum absorption. Maybe the decay was also due to the formation of **MNBOD·Na**, a strong base weak acid buffer.

The DEPT-135 (distortionless enhancement by polarization transfer) experiment is an approach to identify $CH_n$ ($n = 0$–3)

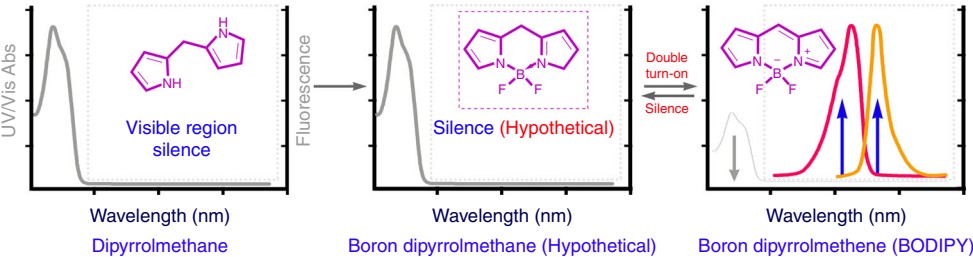

**Fig. 1** Hypothetical chemical transformation induced dual signal double turn on process. The boron dipyrrolmethane has neither absorption (colorless) nor fluorescence in visible region, thus be in a silent state in visible region. But the BODIPY has both strong absorption and fluorescence in visible region. The transforming from dipyrrolmethane to BODIPY accordingly induces sharp photophysical changes with "dual signal" "double turn-on" process

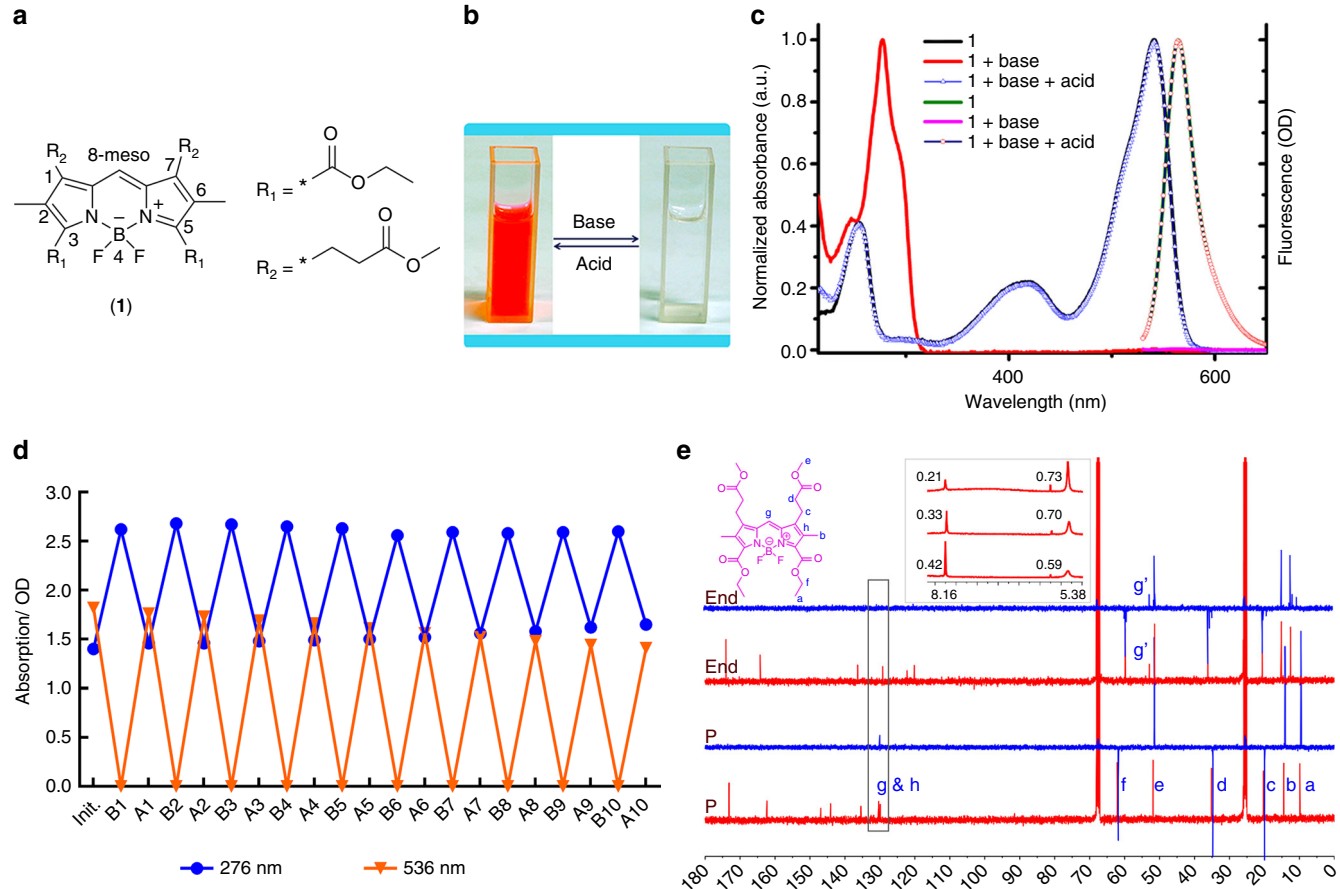

**Fig. 2** The title compound and its properties mediated by base and acid. **a** Structures of **MNBOD 1**, the charge distribution and IUPAC numbering are denoted. **b** Cyclic color change mediated by base and acid. **c** Normalized UV-visible and fluorescence spectra of **1** ethanol solution ($10^{-4}$ M) mediated by $NH_2NH_2$ and HCl. **d** The cycle experiments of **1**. A solution of $10^{-4}$ M **1** in ethanol solution is tuned by 0.1 M NaOH ethanol solution and 1 M HCl aqueous solution alternately and, the absorption at 276 nm (blue) and 536 nm (orange) were recorded simultaneously. **e** The $^{13}$C-NMR and DEPT-135 titrations with hydrazine hydrate. 'P' and 'End' represent pure and colorless state, respectively; blue, DEPT-135 and red $^{13}$C-NMR spectra, respectively. Inset: $^{1}$H-NMR titration of **1** with hydrazine hydrate in deuterated DMSO

carbons where methyl ($CH_3$) and methine (CH) peaks are phased in the positive direction while methylene ($CH_2$) phased in the negative[29]. The DEPT-135 experiments showed that the *meso*-C (be denoted as "g" in Fig. 2e) of **1** shifted upfield from δ + 129.83 p.p.m. to δ + 52.73 p.p.m. with no phase change after addition of hydrazine or ethanediamine. Meanwhile, the $^{1}$H-NMR titrations showed no total integral area change but only the upfield shift of *meso*-H from δ 8.16 to δ 5.38 p.p.m.; in addition, the integral area at δ 5.38 p.p.m. was dynamically complementary to δ 8.16 p.p.m. as one proton and was also proportional to the base added (Fig. 2e inset, and Supplementary Figs. 2–5). The DEPT-135 together with $^{1}$H-NMR titration experiments indicated a possible hybridization change of the *meso*-C from −CH = ($sp^2$) to > CH− ($sp^3$) with no protonation or deprotonation. The possible hybridization change in NMR titrations together with the reversible large spectrophotometric change indicates possible breaking and reconstruction of the chromophore's conjugation.

To provide more direct and solid evidence, single crystals appropriate for X-ray crystallographic analysis were obtained by slow evaporation of THF solutions of **1** with and without hydrazine hydrate. The crystal structures of the colorless and red species are shown in Fig. 3a, b, the crystal data and structure refinement for **1** and **D1** + $NH_2NH_2$ are listed in Supplementary Table 2. For the colorless species (Fig. 3a), two **MNBOD**s are linked together by a single C–C bond (C9–C38, 1.58 Å) at the

*meso*-position. In each **MNBOD**, the bond lengths adjacent to *meso*-C, e.g., C(8)–C(9) and C(9)–C(10), elongate from 1.38–1.39 Å to 1.50–1.51 Å (Table 1), which is in the range of single bond length ($sp^3$–$sp^3$: 1.54 Å; $sp^3$–$sp^2$: 1.51 Å; $sp^3$–$sp$: 1.46 Å; aromatic: 1.39 Å)[30]. Meanwhile, the six-membered B–N rings were no longer coplanar with torsion angles such as N(1)–C(8)-C(9)–C(10) up to 44.5° whereas the torsion angle for the red species such as N(2)–C(8)–C(9)–C(10) was −1° (Supplementary Table 3). Accordingly, the planarity between the two pyrroles of each **MNBOD** was destroyed with the dihedral angle between the two pyrroles increased to 59.2°, while the dihedral angle for red was 10.6° (Fig. 3b, insets). The crystallographic results clearly confirm the formation of a boron dipyrromethane dimer (abbreviated as **D1** hereafter) with the change of meso-C hybridization from $sp^2$ to $sp^3$. The molecular structures of monomer and dimer with critical bond lengths and angles are diagrammed in Fig. 3c. The hybridization change broke the conjugation, thus making the dye colorless and non-fluorescent, silencing the dye in the visible region as a result.

The ESI-MS spectra of both **D1** crystal methanol solution and hydrazine hydrate freshly treated colorless **1** methanol solution showed dimer ions with *m/z* 1073.4 (Supplementary Table 4, Supplementary Figs. 6 and 7). The MS combined with the NMR experiments indicate that, in the solution state, the colorless dimer also formed by self-addition of **1**. The dimerization process both in solution and solid state clearly explains the change of

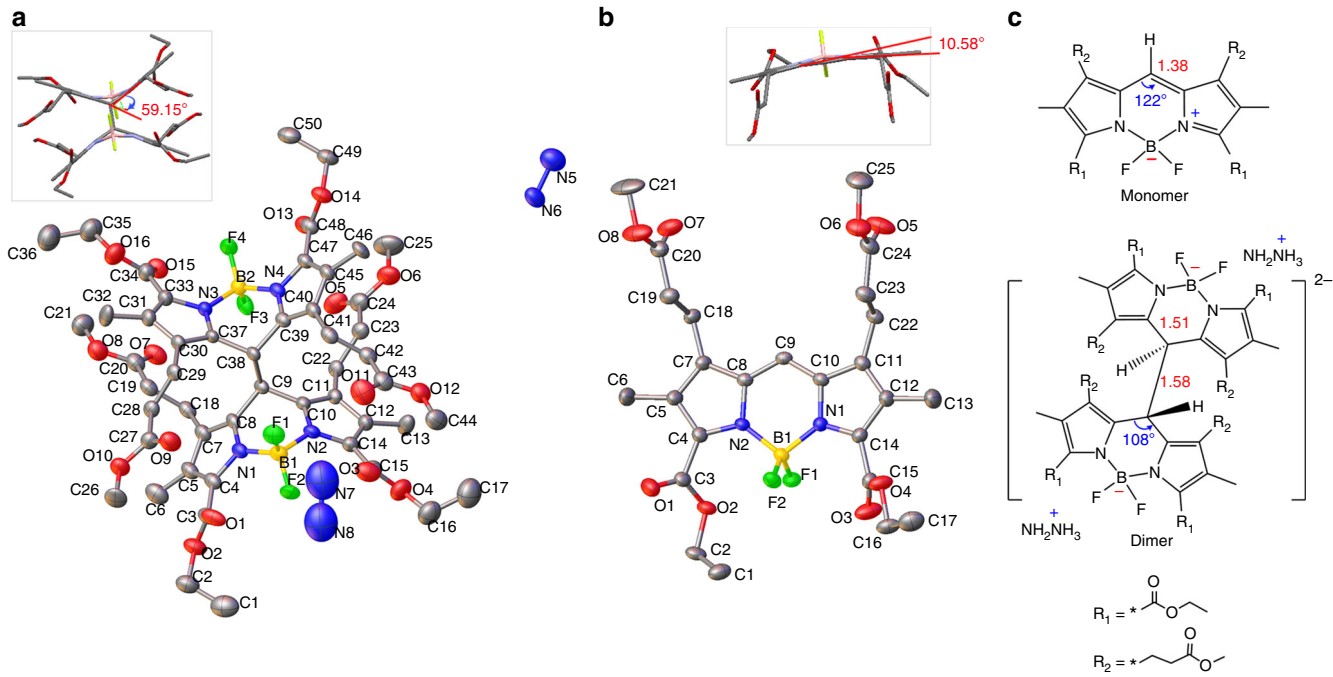

**Fig. 3** Crystal structures of dimer and monomer of **1**. **a** Colorless **D1** and **b** red **1** with thermal ellipsoids at the 50% probability level. Atom numbering schemes are shown; the hydrogen atoms are omitted for clarity. **c** The molecular structures of monomer and dimer with critical bond lengths (red) and angles (blue) denoted

| Table 1 Selected bond lengths and angles for [D1] and [1] | | | |
|---|---|---|---|
| **Bond lengths of D1 [Å]** | | **Bond lengths of 1 [Å]** | |
| C(8)-C(9) | 1.504 | C(8)-C(9) | 1.391 |
| C(9)-C(10) | 1.507 | C(9)-C(10) | 1.379 |
| C(37)-C(38) | 1.510 | | |
| C(38)-C(39) | 1.518 | | |
| C(9)-C(38) | 1.577 | | |
| **Bond angles of D1 [deg]** | | **Bond angles of 1 [deg]** | |
| C(8)-C(9)-C(10) | 107.4 | C(10)-C(9)-C(8) | 121.9 |
| C(37)-C(38)-C(39) | 108.0 | | |

optical properties: the breaking of the chromophore conjugation simultaneously silenced the absorption and fluorescence in the long wavelength region and thus resulted in a clear background.

**Molecular mechanism and calculations**. For further exploration, reference compounds **2** and **3** (Fig. 4a) were synthesized. Investigations showed that **2** had the same properties as compound **1**, while **3** did not show the phenomena when bases were added, indicating that the 3,5-EWGs are essential for transformation. Cyclic voltammetry and theoretical calculation were performed to evaluate the significance of EWG modification. Electron Spin Resonance (ESR) spectra were used to study the mechanism of dimerization.

Cyclic voltammetry (CV) is an important method for the study of electron transfer reactions. Studies on the reduction by CV were performed to give insight into the behavior of **1** and evaluate the electronic interactions between EWGs and the BODIPY core. As shown in Fig. 4b, a nonreversible redox wave was observed at −0.89 V vs. Fc/Fc$^+$ for **1**, while a quasi-reversible ($i_{p,a}/i_{p,c}$ = 0.95) and more negative redox wave for **3** was observed at −1.45 V vs. Fc/Fc$^+$. One can account for this electronic diversity due to EWG-induced increased electron deficiency to **1**, and thus **1** was much easier to reduce than **3**, consequently resulting in a less negative

potential (0.57 V more anodic). Similar results of a cyano group-substituted BODIPY have been reported[31].

We speculated a subsequent dimerization of radical anion [**1**$^{·-}$] that makes the redox wave nonreversible for **1**, whereas fewer dimerizations results in a quasi-reversible redox wave for **3** (Eq. 1 & 2). Consequently, the ESR experiments were employed to detect any possible radicals. It was found that the ESR signal enhanced with base addition (Fig. 4c; for EDA titration, see Supplementary Fig. 8) while no ESR signals were detected for the pure monomer and dimer, indicating that the dimerization process involves free radicals. All ESR experiments were performed in the presence of the diamagnetic spin trap PBN (α-phenyl-N-tert-butylnitrone) otherwise no ESR signal could be detected. The results with PBN suggested the test chemical radicals involved were short-lived. Furthermore, after ESR titration (**1**:Base = 1:1), MS experiments were performed to detect the radical adducts. An ion signal at m/z 735.1 was found and ascribed to [**1**-PBN + Na]$^+$. The MS experiment indicated that the short-lived radical captured by PBN was **1** radical. Thus, the transformation was conclusively confirmed as a radical reaction process that involves the transient free radical of **MNBOD**. The mechanism is predicted as Fig. 4d.

$$2\,\mathbf{1} \underset{\phantom{Base/Red}}{\overset{\text{Base /Red}}{\rightleftharpoons}} 2\,\mathbf{1}^{·-} \rightleftharpoons \mathbf{D1}^{2-} \quad (1)$$
Non-reversible redox wave

$$2\,\mathbf{3} \underset{\text{Ox}}{\overset{\text{Base /Red}}{\rightleftharpoons}} 2\,\mathbf{3}^{·-} \rightleftharpoons \mathbf{D3}^{2-} \quad (2)$$
Quasi-reversible redox wave

Compound **1** is initially reduced by hydrazine via an intermolecular electron transfer process and forms complex [A][32] in which the electron distribution of **1** can be depicted as structure [E] in Fig. 4d. This is actually a radical-anion with negative charge at the boron atom and the free radical localized at

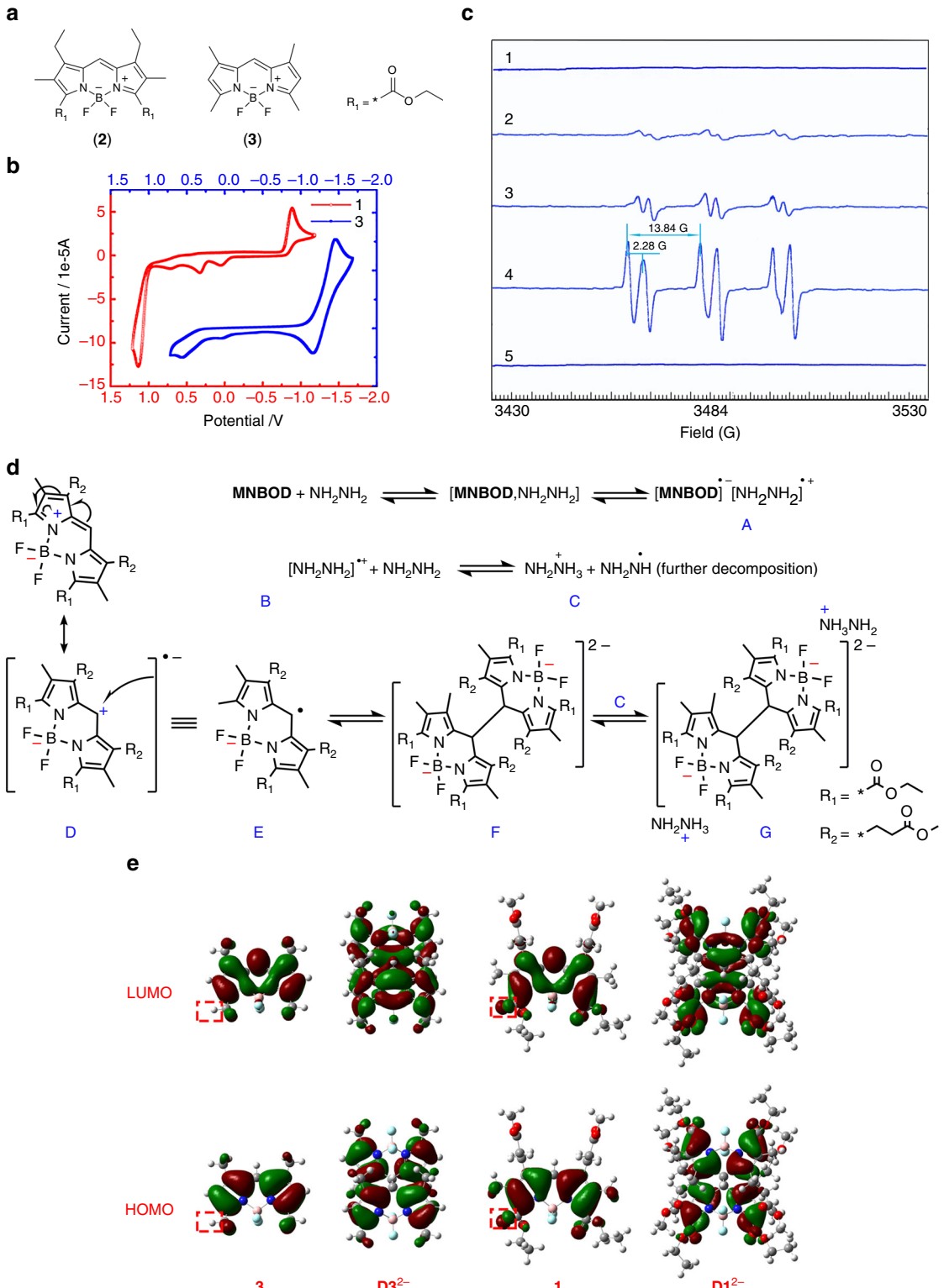

**Fig. 4** The molecular mechanisms. **a** Reference compounds of **2** and **3**. **b** Cyclic voltammetry (CV) of 1 mM **1** (red) and 1 mM **3** (blue). The mechanism for CV redox and dimerization processes is prosed in Eqs. (1) and (2). **c** The ESR titrations with hydrazine hydrate. 1 to 4 denote the concentration change of base (hydrazine hydrate) added. 1, the initial state; 5, the colorless state. Field [G]: 3430-3530, Center Field 3484.00, Sweep width 100.0[G]; $g = 2.00826$; Spin Trap PBN. **d** The mechanism for hydrazine catalyzed dimerization process. **MNBOD** is reduced to a radical-anion (D/E) by hydrazine through the intermolecular electron transfer process. The radical-anions E are readily combine into the dianion F while the hydrazine cation (B) can be further reduced in to hydrazine cation (C). The dimer (F) and (C) combine into the final complex (G). **e** Calculated frontier molecular orbitals of **3**, **D3**$^{2-}$, **1** and **D1**$^{2-}$, HOMO and LUMO orbitals are all obtained by DFT calculations

**Table 2 Electrochemical properties and DFT calculated results of 1, 3 and D1$^{2-}$**

| Compd | $E^{ox,a}$/V | $E^{red,a}$/V | Experimental/eV | | | Theoretical$^e$/eV | | |
|---|---|---|---|---|---|---|---|---|
| | | | HOMO$^b$ | LUMO$^c$ | $\Delta E^d$ | HOMO | LUMO | Energy gap |
| **1** | 1.14 | −0.89 | −5.84 | −3.68 | 2.16 | −5.89 | −3.06 | 2.83 |
| **3** | 0.57 | −1.45 | −5.68 | −3.34 | 2.34 | −5.21 | −2.28 | 2.93 |
| **D1$^{2-}$** | 0.05 | / | / | / | 4.51 | −0.33 | 4.14 | 4.47 |

$^a$Obtained from the potential at peak of oxidation and reduction. $^b$Calculated from HOMO = LUMO−$\Delta E^d$. $^c$Calculated using the empirical LUMO = −(4.44 + $E^{red}_{onset}$). $^d$Estimated from the onset of normalized absorption spectra. Cyclic voltammetry measured using a glassy carbon electrode as a working electrode, a platinum rod as a counter electrode, and Ag/AgNO$_3$ as a reference electrode in CH$_3$CN containing 0.1 M n-Bu$_4$NPF$_6$ as a supporting electrolyte at a scan rate of 100 mV/s under argon atmosphere. $^e$Obtained from the DFT results using Gaussian 09 at the B3LYP/6-31 G(d,p) level.

the *meso*-position. With increasing hydrazine content, the radical cation [B] is reduced into the hydrazine cation [C], followed by further decomposition, just as the processes reported in the literature[33]. At the same time, two molecules of [E] are expected to recombine into the dianion [F]. The dimer [F] complexes with two cations [C], resulting in the final compound [G] as seen in the single X-ray crystal structure.

To provide an insight into the properties of the **MNBOD**, we performed first-principle calculations within the density functional theory (DFT) framework for compounds **1**, **3**, **D1$^{2-}$** and model dimer **D3$^{2-}$**. The geometry of each compound was optimized at the B3LYP/6-31 G(d,p) level with Gaussian 09 package[34], and the optimized geometries with illustration of frontier orbitals of each compound are shown in Fig. 4e and Supplementary Fig. 9. It is worth to note that the calculated geometries of **1** and **D1$^{2-}$** are very close to the obtained single X-ray crystal results (Supplementary Table 5). Hence, the calculated structures of **3** and **D3$^{2-}$** are probably reasonable. The HOMO and LUMO energy levels in a vacuum were estimated and the calculated HOMO, LUMO levels and gaps for compounds **1**, **3** and **D1$^{2-}$** are listed in Table 2. The good correlations between electrochemical studies and DFT calculations demonstrate the validity of these calculations, as shown in Table 2. With EWG modification, **1** had an energy decrease in HOMO, LUMO values and band gap compared with **3**, due to the electron-withdrawing nature of the side groups. The lower LUMO energy equates to a lower reduction potential and higher reactivity.

As shown in Fig. 4e, with EWG substitution, the electrons in **1** delocalize to the side carbonyl groups at the 3,5-position, not only on pyrrole units as in **3**. The expanded delocalization substantially decreases the energy and stabilizes the LUMO state so that the dimerization can easily occur, a consequent effect of which is the nonreversible redox wave of **1** in the CV experiments (Fig. 4b, red traces and Eq. 1). The energy for dimerization as depicted in Eq. (3) was estimated to be −311 kJ/mol (solvation energy ignored), which is within the scale of C–C bond energies (−346 kJ/mol). However, Eq. (4) was estimated at −70 kJ/mol, which suggests that dimer **3** is not stable without EWG modification; conversely, the redox wave for **3** in CV experiments was quasi-reversible (Fig. 4b, blue traces & Eq. 2). This is confirmed from the reported discolor BODIPYs[35,36].

$$1 + 1 \rightarrow D1^{2-} \qquad \Delta E = -311.01 \, \text{kJ/mol} \qquad (3)$$

$$3 + 3 \rightarrow D3^{2-} \qquad \Delta E = -70.30 \, \text{kJ/mol} \qquad (4)$$

In this study, we can draw a conclusion that the EWG modification facilitates the reduction of **MNBOD** (strong absorption, high fluorescence state) into the free radical anion, which readily combined into the metastable dimer dianion, a colorless and non-fluorescent boron dipyrrolmethane. That processes with dramatic photophysical changes gives

this system great potential for a variety of applications. The following experiments were performed to assess the application potentials.

**Advantages for application**. This metastable system has attributes attractive for application. This system has a large absorption spectral shift of 260 nm (280 nm to 540 nm). To our knowledge, no photophysical process has such a large spectral shift. This distinctive phenomenon with its intrinsic strong absorption in the visible region can be utilized for colorimetric or even naked eye probing for rapid, labour saving detection. While for fluorescent approaches, on the other hand, with no absorption in the visible region for the silence state (**D1**), self-absorption can be neglected, unlike other fluorescent probes. This finding is observed because the absorption intensity does not depend on the concentration of **D1** itself but on the concentration of **1** transformed from **D1** dimer. Likewise, the concentration quenching (notably aggregation-caused quenching effects) can also be ignored. This means a higher **D1** concentration can be used to increase the response speed and sensitivity. Although AIE materials can be opted to overcome concentration quenching[37], these materials are sometimes not so sensitive at lower concentration and often have emission in the short wavelength region and, moreover, the self-absorption cannot be avoided for high concentration purposes. This high concentration model makes trace detection simple and efficient. And unlike other "turn-on" probes having only fluorescent response but no simultaneous sharp colorimetric change or vice versa, the probe reported here can be used as a dual-signal probe with color "off-on" accompanying fluorescence "turn-on" process. The "dual signal", "double turn-on" model has advantages over the other turn-on probes, as it supplies a means of mutual calibration. Furthermore, these **MNBODs** have high fluorescent quantum yields both in solids and in solution, with absolute quantum yields of 0.57 and 0.49 for compounds **1** and **2** in CH$_2$Cl$_2$ solution, respectively, and 0.27 and 0.10 for **1** and **2** in solid states, respectively. The emission turn-on process coupled with the high concentration model and high quantum yield luminescence makes **MNBODs** quite sensitive for fluorescent detection because it results in a high turn-on ratio. It provides a distinct competitive advantage in regard to the extremely high turn-on ratio (over 120,000 for 10$^{-4}$ M ethanol solution with NH$_2$NH$_2$/HCl mediation, Fig. 5a), which is critical for low-abundance target detection or super-resolution imaging[5,38]. Finally, unlike one-off reaction-based probes[3,8], the interruption and reversion of conjugation process is reversible and may be used repeatedly; the reversible reaction-based strategy has long been ignored[8,39].

In this study, we hypothesized that the counterion in the complex plays a decisive role in stabilizing the dimer; thus, the sacrifice of the counterion or decomposition of the complex can break the dimer into the **MNBOD** monomer and realize the turn-on process (as depicted in Eq. 5). The $^1$H-NMR experiments (Fig. 5b) together with UV-Fl experiments (Fig. 2c) indicate that the colorless dimer can be induced back to the original monomer

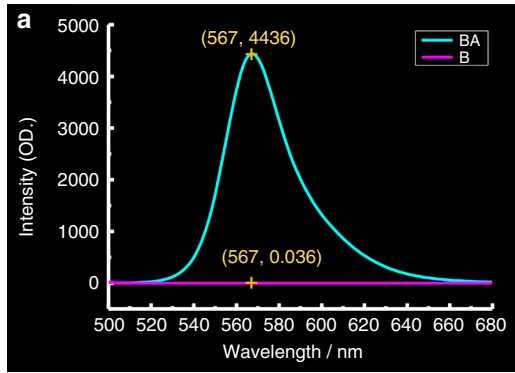
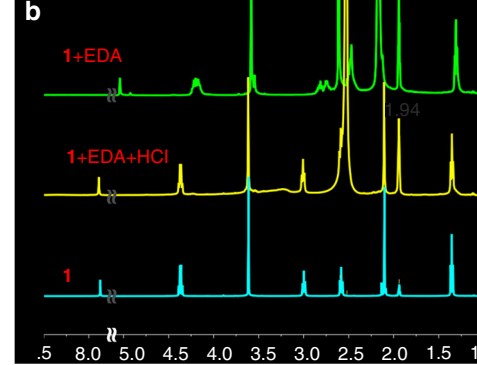

**Fig. 5** High turn-on ration and $^1$H-NMR recover results. **a** A concentration of $10^{-4}$ M **1** ethanol solution mediated with $NH_2NH_2$/HCl; BA denotes adding acid to base treated (denoted as B) colorless solution; S/N = 4436/0.036. **b** $^1$H-NMR experiments, **1** + EDA + HCl denotes add HCl to the colorless **1** + EDA solution; δ 2.56 is ascribed to $-CH_2CH_2-$ of EDA or EDA-HCl, δ 1.94 is $CD_3CN$ solvent peak

state after the addition of acid, which means that the probing process can be achieved while keep the dye intact. In other words, one can choose the appropriate counterion for a specific analyte instead of sophisticated probe design and synthesis. This property is a forte of this system and a means to address the disadvantages of reversible reaction-based probes such as low sensitivity, low selectivity and slow response speed. The sensitivity, selectivity and response speed for sensing depends on the interaction between target analyte and selected counterion. For instance, for the detection of formaldehyde, we chose hydrazine as the counterion to increase the sensitivity, while for application in polymer, we selected ethylenediamine (EDA) to keep the system stable and uniform.

$$D1^{2-}(BH^+)_2 \rightleftharpoons D1^{2-} + 2BH^+ \underset{+ \ B \ / \ cool}{\overset{- \ BH^+/\triangle}{\rightleftharpoons}} 2MNBOD \quad (5)$$

**(Complex, OFF)    (Dimer, OFF)            (Monomer, ON)**

The shortcoming of this system should not, however, be ignored. First, although the turn-on ratio is quite high, the turn-on ratio is base-dependent. For hydrazine, the ratio was approximately 120,000; for EDA, the ratio was approximately 50,000. Second, the counterions were limited to base materials, which means the probing should be confined to base-sensitive analytes.

Hydrazine or its analogs can react with aldehydes readily to form Schiff bases, and this property has long been realized in aldehyde detection[40,41]. We tried to take advantage of the **D1**-$NH_2NH_2$ system to detect formaldehyde, a well-known indoor pollutant that is classified as a carcinogen by US EPA. With the distinctive properties and high reactivity of $NH_2NH_2$, this system has advantages, such as high sensitivity and fast response speed, over the commonly used methods (Fig. 6a).

With the clear background, high turn-on ratio and high concentration model, one can get higher signal to noise ratio (S/N) even for low concentrations. As a result, with a $10^{-4}$ M **D1**-$NH_2NH_2$ ethanol solution we could detect 0.1 ppb of formaldehyde within 10 s and keep a relative high signal to noise ratio (S/N = 23). The World Health Organization's safe-exposure standard for formaldehyde is 80 ppb averaged over 30 min[42]. A good linear correlation ($R^2 = 0.9906$) between the fluorescence intensity and formaldehyde concentration was also found (Fig. 6a, inset). To the best of our knowledge, no such sensitive fluorescent formaldehyde probe has been reported.

The drawback of this system is the low selectivity of hydrazine-based formaldehyde detection methods[40]. The acidity or high

concentration of $CO_2$ may affect experimental results. Our experiments were, however, performed under ambient conditions without special treatment. The solution with formaldehyde can become non-fluorescent once again after the addition of hydrazine, but there is a practical problem with precise control of the amount of hydrazine added in the ppb level; the excessive hydrazine will affect the accuracy and limit of detection in the next cycle, so the repeated use of this system for trace detection is not so convenient.

Characterization of temperature under various conditions requires different approaches. It is great challenge for consecutive probing of the large area or gradient temperatures with traditional thermometers[43,44]. The luminescence methods supply intuitive and remote monitoring approaches for in situ testing of a large area or instant testing of fluidic samples with high spatial resolutions[45]. In this study, we developed a liquid thermometer with the **D1**-$NH_2NH_2$ complex.

The ethanol solution of **MNBOD** with hydrazine hydrate in a quartz NMR tube was sensitive and reversible. One end of the tube was dipped into liquid nitrogen and heated the other end to 70 °C, and then photos were quickly taken under UV-365 irradiation and no irradiation, respectively (Fig. 6b). The temperature-dependent spectra were transformed to Commission Internationale de L'Eclairage (CIE) 1931 coordinates. Figure 6c depicts the gradient color change in a CIE($x,y$) chromaticity diagram versus different temperatures. For the non-irradiated one, the coordinates shifted from red to near white with a wide range along the blue track. While for the emission chromaticity diagram, the coordinates linearly shifted from orange to near yellow first, along the line from red to green, and then turned to the blue region at low temperature. The yellow and blue are a pair of complementary colors, thus the coordinates span a rare wide range in the CIE($x,y$) diagram. The blue fluorescence was due to the emission of **D1**-$NH_2NH_2$ at 400 nm, which tails to the visible region (Fig. 6d).

Even though, for the same temperature, different modes resulted in different coordinates, this approach supplies a way to calibrate and make the results more accurate. For example, it would be troublesome to calculate the temperature near the inflexion points around coordinates 4–6 on emission track (black). The colorimetric track (blue) in this condition will be more reliable, however, as there is a relatively better linearity for the coordinates from 3 to 6 on this track, which just covers that inflexion point of the emission track. For lower temperatures from 6 to 11, the emission track overwhelms the colorimetric one with a much higher resolution (~4 fold). Accordingly, combination of the two complementary models will cover a larger temperature scale and benefit the precision. It is possible to use

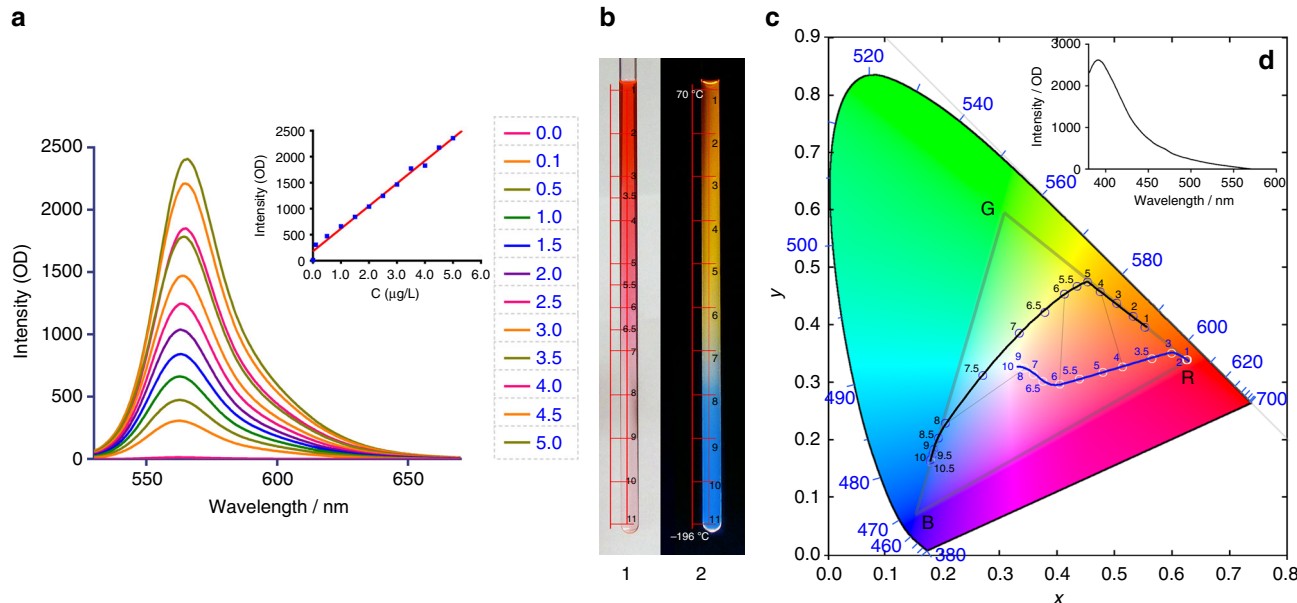

**Fig. 6** Function as formaldehyde and temperature detectors. **a** Formaldehyde detection by colorless **D1**-NH$_2$NH$_2$ fluorescence turn-on process. Fluorescence change of **D1**-NH$_2$NH$_2$ complex (EtOH solution, 10$^{-4}$ M) upon gradually addition of formaldehyde in EtOH (0, 0.1, 0.5, 1.0, 1.5, 2.0, 2.5, 3.0, 3.5, 4.0, 4.5, 5.0 μg/L; 0.1 μg/L = 3.34 × 10$^{-9}$ mol/L, 0.1 ppb) at room temperature. $Y = 435.0 × X + 181.8$ ($R^2 = 0.9906$). The spectra were measured 1 min after formaldehyde addition to minimize relative error, although negligible differentials could be detected between 10 sec. and 1 min. **b** Photographs of gradient colorimetric and fluorescence change of **D1**-NH$_2$NH$_2$ ethanol solution in quartz NMR tube with gradient temperature. 1, gradient temperature, no irradiation; 2, gradient temperature, 365 nm irradiation. **c** The CIE chromaticity diagram with temperature dependence of the (x, y) chromatic coordinates of **D1**-NH$_2$NH$_2$. **d** The fluorescence spectrum of **D1**-NH$_2$NH$_2$ (10$^{-4}$ M) in EtOH solution. Excitation wavelength: 365 nm

**Fig. 7** Synthetic route for compounds **1**, **2** and **3**. The IUPAC numbering system for BODIPY dyes is denoted as (**1**)

such a system with a CCD camera and a CIE resolution software to monitor the gradient temperature instantly and remotely. Unfortunately, we have yet to accomplish the temperature measurement for each point precisely, especially for the lower temperature range, and therefore have not mapped the CIE precisely. This dilemma just emphasizes the urgent need for approaches to monitor the gradient temperature change in fluidic samples. Such work is under study and will be reported in due course.

## Discussion

Chromophore-reaction will bring much larger spectra changes than photophysical processes. The BODIPY is a very efficient dye with strong absorption and high fluorescence. However, with generally accepted chemical robustness and environmental insensitivity, there are no reports about using the structure change of the BODIPY chromophore.

We designed and synthesized **MNBOD** with EWGs around the conjugation core and without substitution at the *meso*-position.

Double EWGs connected to the 3,5-positions make the **MNBOD** core more electron deficient and more active than typical BOD-IPYs. Under the base condition, two **MNBOD** monomers link up reversibly through a free radical reaction to form a metastable boron dipyrromethane dimer. The reversible transformation between the conjugated and non-conjugated states leads to a synchronous colorimetric and fluorescent double turn-on process with "low background" and "high turn-on ratio". Such a system can be utilized as chromophore-reaction-based, dual-signal, high sensitivity probes.

In addition, we have provided a fundamental comprehension of this unique phenomenon which could shed new light on the design of novel BODIPY fluorescent probes with desirable properties and prospective application.

## Methods

**General methods**. All organic solvents were commercially available, dried and distilled by appropriate methods before use.

Electrospray ionization mass spectra (ESI-MS) were recorded on Waters Micromass Q-TOF mass spectrometer, MALDI-TOF MS measurements were carried out on a Bruker BIFLEX III spectrometer. Samples for C, H and N analyses were dried under vacuum and the analysis was performed with a Carlo Erba-1106 elemental analyser. $^1$H, $^{13}$C NMR spectra were recorded on Bruker dmx AVANCE 400 MHz, 600 MHz NMR spectrometers at room temperature with CDCl$_3$/TDF/ DMSO/CD$_3$CN as solvent, chemical shifts are referenced to residual solvent peaks with respect to TMS = δ 0 ppm. Experimental DEPT spectra were recorded on Bruker dmx AVANCE 600 MHz NMR spectrometers for proton with flip angle of the final I-spin pulse (β) set to 3π/4.

UV-vis absorption spectra were recorded on a Hitachi U-3010 spectrophotometer. Fluorescence spectra were recorded on a Hitachi F-4500 fluorescence spectrophotometer. Fluorescence quantum yields (Φ$_f$) were determined by the absolute method on HAMAMATSU Absolute PL Quantum Yield Spectrometer C11347.

**X-ray crystallography**. Accurate unit cell parameters were determined by a least-squares fit of 2θ values, and intensity data sets were measured on Rigaku Raxis Rapid IP diffractometer with Cu-Kα radiation (λ = 0.71073 Å). The intensities were corrected for Lorentz and polarization effects, but no corrections for extinction were made. All structures were solved by direct methods. The non-hydrogen atoms were located in successive difference Fourier synthesis. The final refinement was performed by full-matrix least-squares methods with anisotropic thermal parameters for non-hydrogen atoms on F$^2$. The hydrogen atoms were added theoretically as riding on the concerned atoms. Crystallographic data for structure analyses are summarized in Supplementary Data set 1 and 2 with selected bond lengths and angles data listed.

**ESR measurements**. ESR spectra were recorded on a Bruker E500 spectrometer in a standard resonance cavity (ST4102). The monomer/dimer and PBN (1 equiv.) were dissolved in 50 μL of organic solvent (for EDA titration, toluene; for hydrazine hydrate titration, THF) with a final concentration of 0.01 M (The solid **1** monomer and **D1**-NH$_2$NH$_2$ crystal were also tested, even though no ESR signal was detected). The samples were flushed with nitrogen (10–15 min) to remove oxygen, and the bases were added quickly under a nitrogen atmosphere just before ESR analysis. The ESR spectra were recorded at room temperature. ESR conditions: microwave frequency, 9.338 GHz; microwave power, 10 mW; modulation amplitude, 10 G; modulation frequency, 100 kHz.

**Theoretical calculations**. First-principle calculations were performed within the density functional theory (DFT) framework for compounds **1**, **3**, **D1**$^{2-}$ dianion and model dimer **D3**$^{2-}$ dianion. The geometry of each compound was optimized at the B3LYP/6-31 G(d,p) level with Gaussian 09 package[34], and the optimized geometry with illustration of frontier orbitals of compounds are shown in Fig. 4e; the HOMO and LUMO energy levels in vacuo were estimated and the calculated HOMO, LUMO levels and gaps for compounds **1**, **3** and **D1**$^{2-}$ are listed in Supplementary Data set 3.

**Synthetic procedures**. The synthetic route for preparation of **MNBOD** (**1**) and reference compounds (**2**) and (**3**) is shown in Fig. 7. The detailed description of the synthesis of related pyrroles (**1b**, **2a**) was presented in references [46–48].

**Diethyl 5,5'-methylenebis(4-(3-methoxy-3-oxopropyl)-3-methyl-1$H$-pyrrole-2-carboxylate) (1b)**. $^1$H NMR (400 MHz, CDCl$_3$) δ 9.14 (s, 2 H), 4.24 (q, J = 7.1 Hz, 4 H), 3.97 (s, 2 H), 3.67 (s, 6 H), 2.77 (t, J = 7.2 Hz, 4 H), 2.51 (t, J = 7.1 Hz, 4 H), 2.28 (s, 6 H), 1.30 (t, J = 7.1 Hz, 6 H). $^{13}$C NMR (400 MHz, CDCl$_3$) δ 173.97, 161.81, 130.50, 126.71, 120.21, 118.53, 59.89, 51.87, 34.67, 22.61, 19.41, 14.66,

10.77. ESI-MS $m/z$ (calc. for C$_{25}$H$_{34}$N$_2$O$_8$): 491.5 [M + H]$^+$ (calc. 491.5), 513.5 [M + Na]$^+$ (calc. 513.5).

**Diethyl 5,5'-methylenebis(4-ethyl-3-methyl-1$H$-pyrrole-2-carboxylate) (2a)**. $^1$H NMR (400 MHz, CDCl$_3$) δ 8.89 (s, 2 H), 4.25 (q, J = 7.1 Hz, 4 H), 3.87 (s, 2 H), 2.41 (q, J = 7.5 Hz, 4 H), 2.28 (s, 6 H), 1.30 (t, J = 7.1 Hz, 6 H), 1.04 (t, J = 7.5 Hz, 6 H). $^{13}$C NMR (400 MHz, CDCl$_3$) δ 162.09, 129.01, 127.13, 124.40, 118.13, 60.04, 23.14, 17.45, 15.66, 14.71, 10.70. ESI-MS $m/z$ (calc. for C$_{21}$H$_{30}$N$_2$O$_4$): 375.3 [M + H]$^+$ (calc. 375.2), 397.3 [M + Na]$^+$ (calc. 397.2).

The title compound (**1**) and reference compound (**2**) were synthesized by the known procedure of BODIPY preparation.

**3,5-bis(Ethoxycarbonyl)-4,4'-difluoro-1,7-bis(3-methoxy-3-oxopropyl)-2,6-dimethyl-8$H$-boron dipyrromethene (1)**. $^1$H NMR (400 MHz, CDCl$_3$) δ 7.73 (s, 1 H), 4.44 (q, J = 7.1 Hz, 4 H), 3.65 (s, 6 H), 2.99 (t, J = 7.3 Hz, 4 H), 2.57 (t, J = 7.3 Hz, 4 H), 2.15 (s, 6 H), 1.43 (t, J = 7.1 Hz, 6 H). $^{13}$C NMR (400 MHz, THF) δ 173.14, 162.32, 146.89, 144.11, 135.83, 130.34, 129.93, 62.11, 51.84, 35.13, 20.28, 14.46, 9.87. ESI-MS $m/z$ (calc. for C$_{25}$H$_{31}$BF$_2$N$_2$O$_8$): 559.3 [M + Na]$^+$ (calc. 559.3), 575.3 [M + K]$^+$ (calc. 575.4). Elemental analysis for calculated for C$_{25}$H$_{31}$BF$_2$N$_2$O$_8$: C, 55.99%; H, 5.83%; N, 5.22%. Found: C, 55.77%; H, 5.81%; N, 5.19%.

**3,5-bis(Ethoxycarbonyl)-1,7-diethyl-4,4'-difluoro-2,6-dimethyl-8$H$-boron dipyrromethene (2)**. $^1$H NMR (400 MHz, CDCl$_3$) δ 7.33 (s, 1 H), 4.44 (q, J = 7.1 Hz, 4 H), 2.64 (q, J = 7.6 Hz, 4 H), 2.13 (s, 6 H), 1.43 (t, J = 7.1 Hz, 6 H), 1.17 (t, J = 7.6 Hz, 6 H). TOF-MS $m/z$ (calc. for C$_{21}$H$_{27}$BF$_2$N$_2$O$_4$): 420.3 [M]$^+$ (calc. 420.26).

**4,4'-Difluoro-1,3,5,7-tetramethyl-8$H$-boron dipyrromethene (3)** was synthesized using the process reported by L. Wu[49]. $^1$H NMR (400 MHz, DMSO-d$_6$) δ 7.03 (s, 1 H), 6.04 (s, 2 H), 2.53 (s, 6 H), 2.24 (s, 6 H). TOF-MS $m/z$ (calc. for C$_{13}$H$_{15}$BF$_2$N$_2$): 248 [M]$^+$ (calc. 248.1).

**Crystal cultivation**. (Monomer **1**): Slow evaporation of 1 mL of a THF solution of **1** (10.7 mg, 0.02 M) resulted in red blocks suitable for single-crystal X-ray diffraction. m.p.: 154.3–155.0 °C, yield: 92%.

(**D1**): Compound **1** (5.4 mg) was dissolved in 1 mL of THF and subsequently treated with hydrazine until the solution became colorless; slow evaporation of that solution resulted in colorless blocks of **D1**-NH$_2$NH$_2$ suitable for single-crystal X-ray diffraction. m.p.: 150.1–151.1 °C, yield: 83.3%, High resolution ESI-MS $m/z$ (calc. for C$_{50}$H$_{62}$B$_2$F$_4$N$_6$O$_{16}$): 1106.4812 [Dimer + hydrazine + 2 H] (calc. 1106.4814), 1073.4435 [Dimer + H]$^+$ (calc. 1073.4361).

**Data availability**. The authors declare that all the relevant data are available within the paper and its Supplementary Information file or from the corresponding author upon reasonable request. The X-ray crystallographic coordinates for **1**, **D1** can be obtained free of charge from The Cambridge Crystallographic Data Centre via www.ccdc.cam.ac.uk/data_request/cif under deposition numbers CCDC 1497162 and 1497164, respectively.

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

## Acknowledgements

This work was supported by the National Basic Research Programme (2013 CB834703, 2013 CB834505) and the National Natural Science Foundation of China (21233011), and the Russian Foundation for Basic Research (13-03-911550-rфEп-а). We thank Dr. Jianming Chen for theoretical calculations. D.H. gives special thanks to his girlfriend Dr. Yingying Chen for valuable advice and manuscript revision.

## Author contributions

D.H. and T.Z. are co-first author. D.H. designed the research, performed the experiments, analyzed the data, wrote and revised the manuscript. T.Z. assisted in data test. S.L. and J.C. performed the calculations, and S.L. also helped revising the manuscript. T.L. helped resolving the single crystal structure. T.Y., X.Z., R.H., J.F., S.W. and Y.L. assisted in CV, TOF, NMR etc. L.N.S. and B.A.T. provided valuable advices for the dimerization mechanism.G.Y. and J.M. supervised the project.

## Additional information

**Competing interests:** The authors declare no competing financial interests.

