## [Peer Review File · Nature Communications]

Reviewers' comments:

Reviewer #1 (Remarks to the Author):

The contribution "Ultrasensitive Reversible Chromophore-Reaction of BODIPY and Its Application as Dual-Signal High Turn-On Ratio Probe" deals with synthesis of meso-naked BODIPY functionalized with the electron-withdrawing groups and study of unusual reversible dimerization of such systems in the presence of bases, such as ethylenediamine or hydrazine. This process has been studied completely by different methods: UV-vis, NMR, XRD, Cyclic voltammetry, ESR, and formation of dimer has been clearly proved. The dimerization process leads to the dramatic change of system's optical properties: vanishing of characteristic adsorption and fluorescence emission of BODIPY after addition of base. Also, the importance of electron-withdrawing groups for this process and possible applications has been discussed. The authors propose the original and excellent study. To my knowledge, no analogical systems have been described. The object of study has been chosen carefully (importance of electron-withdrawing groups) and thoroughly studied – from rational design to application. The manuscript is clearly written and could be interesting for not only the specialists in this field but also for all chemical community. The claims are convincing and very explicative, that makes manuscript accessible to nonspecialists. So, I recommend to accept this article for publication

Reviewer #2 (Remarks to the Author):

In this manuscript authors designed and characterized a meso-naked BODIPY (MNBOD) with two electron-withdrawing groups around the BODIPY core. The obtained system reported to be highly sensitive to bases that caused diminishing of fluorescence and color and obtained dimeric BODIPY derivatives via radicalic reaction. Also they successfully restored the fluorescence by addition of an acid or formaldehyde. In my opinion, the content of this manuscript is really interesting, the research topic is timely and of great importance to a wide research community.

However, I m not convinced about the probe applications. First authors only used HCl or formaldehyde as an acid. The purity of the HCl (aq or pure) was not mentioned. Did they use HCl in water? If so, water was not taken into account in mechanism and probe preparation. Also, as base authors only worked with hydrazine and ethyl amine. I think they might be consider working with other acids and bases (not amine based for example) since they mentioned the effect of counter ion and the influence of various acidic/ basic species.

The system proposed as reversible probe and can be cycled. Is it possible working on the kinetic mode to see cycle number? I have concerns about the effect of the temperature on formaldehyde probe. Since authors also proposed the D1-NH₂NH₂ system as a temperature probe, temperature dependence may affect the accuracy of formaldehyde probe in practice.

Consequently, I do not recommend publication of the manuscript in the Nature Communications

Reviewer #3 (Remarks to the Author):

This manuscript manuscript entailed "Ultrasensitive Reversible Chromophore-Reaction of BODIPY and Its Application as Dual-Signal, High Turn-On Ratio Probe" can be regarded as one of the top-notch papers in the fields of BODIPYs as well as molecular sensors. The authors have introduced the very basics of the properties of BODIPYs to the readers, and through experimental excellence, unveiled a number of chemical and physical phenomenon related to the newly designed BODIPY dyes. Although the idea is quite simple, its simplicity should attract a broad readership. Most significantly, the radical chemistry related to the meso-position of the BODIPY dye is of fundamental importance, and might be retroactively able to relate to many other previous studies where this phenomenon might have been overlooked. With the presentation of the crystal structures, the authors have put an undeniable basis to their claims. This is the most interesting prospect of this manuscript, as all of the claims have been soundly proven by the experiments. I also admire the high quality graphical presentation of the data throughout the manuscript.

However, I would suggest the following changes

- (1) Chart 1: The hypothetical compound is chemically impossible. It should be an anionic species. Please remove the hypothetical portion. If the data are not experimental results, please present the concept in a concise graphics, excluding the data.
- (2) Chart 3: Please clean up the structures. They appear distorted.
- (3) In Figure 5, the G value (x-axis) should be given.
- (4) Please revise the English used throughout the manuscript. Please avoid using terms like 'fluke' etc.
- (5) Please discuss the symmetry elements for the DFT obtained structures, compared to X-ray diffraction obtained structures. Were the puckered observed structures taken into considerations? DFT often fails to describe the puckering of BODIPY dyes, please add a detailed description of the theoretical investigation compared to the experimental findings.
- (6) Investigations of the fluorescence turn-on phenomena inside living cells should be investigated to increase the impact of the paper.
- (7) In chart 1. Inside right figure change 'visable regin' to visible region?

Dear Reviewers,

Thank you for the time and effort you've put into reviewing the previous version of our manuscript entitled "Ultrasensitive Reversible Chromophore-Reaction of BODIPY and Its Application as Dual-Signal, High Turn-On Ratio Probe" (ID: NCOMMS-17-11742). Your comments are valuable and significant to our researches which enabled us to revise and improve our paper. We have studied all your comments carefully and made corrections to our manuscript. Accordingly, we have uploaded a copy of the original manuscript with all the changes highlighted by using the track changes mode in MS Word.

Appended to this letter is a list of main corrections in the paper and our point-by-point response to your comments. The manuscript has been polished up by native English speakers.

Thank you for you kind work.

Sincerely yours,

The Authors

Main corrections:

1. Redraw Chart 1, changed the hypothetic compound to a complex and revise the "Visible Regin" to "Visible Region" as Reviewer 3 suggested.
 2. We used the NaOH and HAc as the base and acid and realized the discolour and recover process and, supplemented the data to the revised manuscript as Reviewer 2 suggested.
 3. We supplemented the cycle experiment with the cycle number to the revised manuscript as Reviewer 2 suggested.
 4. Redraw Figure 6 and added the G value to the x-axis as Reviewer 3 suggested.
 5. Added a description comparing the calculated structure and crystal structure as Reviewer 3 suggested.
 6. The manuscript has been polished up by native English speakers and we've uploaded the certificate as SI materials.
-

Reviewer #1 (Remarks to the Author):

The contribution "Ultrasensitive Reversible Chromophore-Reaction of BODIPY and Its Application as Dual-Signal High Turn-On Ratio Probe" deals with synthesis of meso-naked BODIPY functionalized with the electron-withdrawing groups and study of unusual reversible dimerization of such systems in the presence of bases, such as ethylenediamine or hydrazine. This process has been studied completely by different methods: UV-vis, NMR, XRD, Cyclic voltammetry, ESR, and formation of dimer has been clearly proved. The dimerization process leads to the dramatic change of system's optical properties: vanishing of characteristic adsorption and fluorescence emission of BODIPY after addition of base. Also, the importance of electron-withdrawing groups for this process and possible applications has been discussed. The authors propose the original and excellent study. To my knowledge, no analogical systems have been described. The object of study has been chosen carefully (importance of electron-withdrawing groups) and thoroughly studied from rational design to application. The manuscript is clearly written and could be interesting for not only the specialists in this field but also for all chemical community. The claims are convincing and very explicative, that makes manuscript accessible to non-specialists. So, I recommend to accept this article for publication.

Reply: Thank you for finding the potential significance in our study.

Reviewer #2 (Remarks to the Author):

In this manuscript authors designed and characterized a meso-naked BODIPY (MNBD) with two electron-withdrawing groups around the BODIPY core. The obtained system reported to be highly sensitive to bases that caused diminishing of fluorescence and color and obtained dimeric BODIPY derivatives via radicalic reaction. Also they successfully restored the fluorescence by addition of an acid or formaldehyde. In my opinion, the content of this manuscript is really interesting, the research topic is timely and of great importance to a wide research community.

However, I m not convinced about the probe applications. First authors only used HCl or formaldehyde as an acid. The purity of the HCl (aq or pure) was not mentioned. Did they use HCl in water? If so, water was not taken into account in mechanism and probe preparation. Also, as base authors only worked with hydrazine and ethyl amine. I think they might be consider working with other acids and bases (not amine based for example) since they mentioned the effect of counter ion and the influence of various acidic/ basic species.

The system proposed as reversible probe and can be cycled. Is it possible working on the kinetic mode to see cycle number? I have concerns about the effect of the temperature on formaldehyde probe. Since authors also proposed the D1-NH₂NH₂ system as a temperature probe, temperature dependence may affect the accuracy of formaldehyde probe in practice.

Consequently, I do not recommend publication of the manuscript in the Nature Communications

1. **Comment:** First authors only used HCl or formaldehyde as an acid.

Reply: For the acid, we used HCl (aq.), formaldehyde (aq.) and acetic acid (HAc, pure organic acid) et al. and the acetic acid actually works (Fig. 1) (See line 101-102).

Fig. 1 The cycle for HAc and NaOH. *Initial*, the original MNBD solution; NaOH, add NaOH to MNBD solution; HAc, add HAc to the colourless MNBD solution mediated by NaOH before.

2. **Comment:** Also, as base authors only worked with hydrazine and ethyl amine. I think they might be consider working with other acids and bases (not amine based for example) since they mentioned the effect of counter ion and the influence of various acidic/ basic species.

3. **Reply:** Thanks for the suggestion. Our system is also sensitive to KOH, NaOH (Fig. 1 above) and NaHCO₃. We've added the KOH to MNBOD solution and got the colourless crystal, in which two K cations complexed with one dimer; that's the way we confirmed that the dimer is a dianion system (two dimers share 4 K atoms, we prepared this results for another article). We can check the crystal structure of MNBOD-K₂ complex through the link below:

<https://www.ccdc.cam.ac.uk/structures/search?access=referee&searchdepnms=1497368&searchauthor=Hu>

4. **Comment:** The purity of the HCl (aq or pure) was not mentioned. Did they use HCl in water? If so, water was not taken into account in mechanism and probe preparation.

Reply: HCl and NaHCO₃ et al. were all tested in aqueous solution (aqueous solution of HCl and aqueous solution of NaHCO₃ added to MNBOD ethanol solution) but led to the contrary results. So the water isn't essential for that processes.

Moreover, it is an advantage that our MNBOD radical-anion doesn't sensitive to the water.

5. **Comment:** I have concerns about the effect of the temperature on formaldehyde probe. Since authors also proposed the D1-NH₂NH₂ system as a temperature probe, temperature dependence may affect the accuracy of formaldehyde probe in practice.

Reply: Yes you are right. However, we can control the experimental temperature for the probe process. The formaldehyde detection process was under room temperature and we got good linear correlation ($R^2=0.9906$) in our reports; if we had and added a temperature control instrument to the fluorescent addition experiments, maybe we should get a better linear correlation.

6. **Comment:** The system proposed as reversible probe and can be cycled. Is it possible working on the kinetic mode to see cycle number?

Reply: Thank you for your constructive advices. We use the absorption experiment to determine the cycle number and the results were supplemented to the revised manuscript. (Line 103-122) The cycle experiment was performed in a 10 mm × 10 mm quartz cell with 2 mL of 10⁻⁴ M MNBOD ethanol solution. 0.1 M of NaOH ethanol solution and 1 M of HCl aqueous solution were used as base and acid. The volume of base and acid added and the subsequent UV absorption at 276 nm and 536 nm for each cycle were listed in Table 1 and 2. From table 2 we can see an average decay of 0.025 for each cycle, which means the absorption at 536 nm decayed to a half initial value at cycle 27. With cycle number increasing, the solution volume will increase notably that decreases the concentration and subsequently decreases the maximum absorption. Maybe it was due to the formation of MNBOD.Na, a strong base weak acid buffer or the accumulation of NaCl in ethanol solution.

Table 1. The volume of base and acid added for each cycle.

Vol. (μL) \ No.	1	2	3	4	5	6	7	8	9	10	Total Vol. (μL)
NaOH (B, 0.1 M)	11	10	9.5	9.3	10	10	11	10	10	10	100.8
HCl (H, 1 M)	8	7.5	8.1	8.8	8.2	8	8	8	8	8	80.6

Table 2. The UV absorption at 276 nm and 536 nm for each cycle.

	276 nm	536 nm	$OD_{H(n+1)}/OD_{Hn}$ at 536nm
Init.	1.40	1.82	--
B1	2.62	0.00	
H1	1.46	1.76	0.97
B2	2.68	0.00	
H2	1.46	1.73	0.98
B3	2.67	0.00	
H3	1.48	1.69	0.98
B4	2.65	0.00	
H4	1.49	1.66	0.98
B5	2.63	0.00	
H5	1.50	1.61	0.97
B6	2.56	0.00	
H6	1.52	1.56	0.97
B7	2.59	0.00	
H7	1.56	1.52	0.97
B8	2.58	0.00	
H8	1.58	1.48	0.97
B9	2.59	0.00	
H9	1.62	1.44	0.98
B10	2.60	0.00	
H10	1.65	1.41	0.98

Fig. 2 The cycle experiments of M MNBOD (10^{-4} , ethanol solution) with 0.1 M NaOH (ethanol solution) and 1 M HCl (aqueous solution).

Thank you for your constructive comments for improving our manuscript. We performed cycle experiments with NaOH and HCl, and incorporated them into our manuscript, in accordance with your suggestions.

Reviewer #3 (Remarks to the Author):

This manuscript manuscript entailed "Ultrasensitive Reversible Chromophore-Reaction of BODIPY and Its Application as Dual-Signal, High Turn-On Ratio Probe" can be regarded as one of the top-notch papers in the fields of BODIPYs as well as molecular sensors. The authors have introduced the very basics of the properties of BODIPYs to the readers, and through experimental excellence, unveiled a number of chemical and physical phenomenon related to the newly designed BODIPY dyes. Although the idea is quite simple, its simplicity should attract a broad readership. Most significantly, the radical chemistry related to the meso-position of the BODIPY dye is of fundamental importance, and might be retroactively able to relate to many other previous studies where this phenomenon might have been overlooked. With the presentation of the crystal structures, the authors have put an undeniable basis to their claims. This is the most interesting prospect of this manuscript, as all of the claims have been soundly proven by the experiments. I also admire the high quality graphical presentation of the data throughout the manuscript.

However, I would suggest the following changes

- (1) Chart 1: The hypothetical compound is chemically impossible. It should be an anionic species. Please remove the hypothetical portion. If the data are not experimental results, please present the concept in a concise graphics, excluding the data.
 - (2) Chart 3: Please clean up the structures. They appear distorted.
 - (3) In Figure 5, the G value (x-axis) should be given.
 - (4) Please revise the English used throughout the manuscript. Please avoid using terms like 'flake' etc.
 - (5) Please discuss the symmetry elements for the DFT obtained structures, compared to X-ray diffraction obtained structures. Were the puckered observed structures taken into considerations? DFT often fails to describe the puckering of BODIPY dyes, please add a detailed description of the theoretical investigation compared to the experimental findings.
 - (6) Investigations of the fluorescence turn-on phenomena inside living cells should be investigated to increase the impact of the paper.
 - (7) In chart 1. Inside right figure change 'visable regin' to visible region?
-

1. **Comment:** Chart 1: The hypothetical compound is chemically impossible. It should be an anionic species. Please remove the hypothetical portion. If the data are not experimental results, please present the concept in a concise graphics, excluding the data.

Reply: Thank you for your precious advices and we deleted the data in the graphics.

The hypothetical compound in last version was little confused, so we redrawn it with an arrow, which means a complex process from F to B atom. Analogous boron dipyrromethane was reported but without absolutely structure characterization (Bellut, H.; Miller, C. D.; Koster, R. Syn. Inorg. Met.-Org. Chem 1971, 1, 83.).

2. **Comment:** Chart 3: Please clean up the structures. They appear distorted.

Reply: Thank you for your advice. We have optimized the graph. However, there are many long side chains that making overlap unavoidable. We've uploaded the crystal structure as SI material, or you can see the crystal structure of MNBOD dimer and monomer in CCDC as links below:

Dimer:

<https://www.ccdc.cam.ac.uk/structures/search?access=referee&searchdepnms=1497164&searauthor=hu>

Monomer:

<https://www.ccdc.cam.ac.uk/structures/search?access=referee&searchdepnms=1497162&searauthor=Hu>

3. **Comment:** In Figure 5, the G value (x-axis) should be given.
Reply: Thank you and we've redrawn that figure with G value as you advised. (Figure 6 in this version.)
4. **Comment:** Please revise the English used throughout the manuscript. Please avoid using terms like 'fluke' etc.
Reply: Thank you, good English is still a challenge for us. And this version of the manuscript has been polished up by native English speakers.
5. **Comment:** Please discuss the symmetry elements for the DFT obtained structures, compared to X-ray diffraction obtained structures. Were the puckered observed structures taken into considerations? DFT often fails to describe the puckering of BODIPY dyes, please add a detailed description of the theoretical investigation compared to the experimental findings.
Reply: Well, it is worth to note that the calculated geometries of **1** and **D1²⁻** are very close to the obtained single X-ray crystal results as show in the table below. So the calculated results are reasonable and quite similar to the crystal structures. (Line 254-257)

D1²⁻			1		
Key structure factors	Calc.	Cryst.	Key structure factors	Calc.	Cryst.
C8-C9 (Length/Å)	1.506	1.504	C(8)-C(9) (Length/Å)	1.393	1.391
C9-C38 (Length/Å)	1.594	1.577	C(9)-C(10) (Length/Å)	1.388	1.379
C8-C9-C10 (Angle/deg)	107.36	107.40	C10-C9-C8 (Angle/deg)	122.74	121.88
C37-C38-C9 (Angle/deg)	112.65	111.31			
N1-C8-C9-C10 (Torsion/deg)	46.03	44.50	N1-C10-C9-C8 (Torsion/deg)	2.85	1.49
Dihedral angle of two pyrroles (deg)	62.61	59.15	Dihedral angle of two pyrroles (deg)	12.25	10.58

6. **Comment:** Investigations of the fluorescence turn-on phenomena inside living cells should be investigated to increase the impact of the paper.
Reply: Thanks for the suggestion. Investigating the turn-on phenomena in living cell should be more attractive, such as to distinguish the cancer cell and the normal cell. But this system in cell is not so sensitive as in the solution. Maybe it was due to the complex buffer system in living cells. We will make some modification to our molecule in the future and try to realize that process.
7. **Comment:** In chart 1. Inside right figure change 'visable regin' to 'visible region'?
Reply: We've corrected that mistake.

Thank you very much for your patience and careful checking. Your evaluation our paper as "one of the top-notch papers in the fields of BODIPYs as well as molecular sensors" is a great encouragement to us. We have improved our paper by addressing your comments, however,

to probe the living cell is still a challenge right now. We need to make more effort to improve our system.

REVIEWERS' COMMENTS:

Reviewer #2 (Remarks to the Author):

The manuscript was revised as suggested and can be accepted for publishing

Reviewer #3 (Remarks to the Author):

The authors have done good job and all my queries are taken care in the revised manuscript. I am happy with the revision and pleased to recommend the manuscript for publication in Nature communications.